# Extended Stanford Type-A Aortic Dissection with Multivessel Coronary and Peripheral Artery Involvement: An Autopsy Case Report

**DOI:** 10.3390/healthcare11030386

**Published:** 2023-01-29

**Authors:** Maria Alexandri, Maria Tsellou, Nikolaos Goutas, Konstantina Galani, Stavroula Papadodima

**Affiliations:** Department of Forensic Medicine and Toxicology, National and Kapodistrian University of Athens, 11527 Athens, Greece

**Keywords:** autopsy, carotid, coronary, iliac, radial, Stanford A aortic dissection, subclavian

## Abstract

We report the case of a 64-year-old male who died suddenly short after his admission to hospital because of strong chest pain and before any clinical diagnosis was established. His medical history included coronary disease with coronary by-pass surgery at the age of 40 years old, uncontrolled hypertension, diabetes mellitus, and elevated levels of cholesterol. The autopsy revealed quite a rare case of Stanford A aortic dissection with extension to the common and internal carotid arteries; the subclavian, axillary, brachial, and radial arteries; three coronary arteries; the superior mesenteric artery; and the iliac arteries. There was no histological evidence of aortitis or connective tissue disease. The death did not result from the rupture of the aortic dissection, but from myocardial ischemia due to coronary occlusion in combination with hemodynamic disturbance from stress caused by the extended aortic dissection.

## 1. Introduction

Aortic dissection is a life-threatening condition that affects 5–30 in 1,000,000 individuals each year [1]. Patients may present with pain and shortness of breath, symptoms that mimic other conditions such as ischemic cardiac disease and perforated gastric ulcer. It is associated with high morbidity and mortality rates and many patients die shortly after the onset of symptoms, before presentation to the emergency department, or before a diagnosis is established [2,3]. Historically, many of the early reports regarding acute aortic dissection were based, to a great extent, on autopsy findings [4]. Since then, several autopsy studies have shown that in a considerable percentage of cases, aortic dissection is first diagnosed during the autopsy, even when patients are under medical care [5,6,7,8].

Aortic dissections are classified anatomically using two systems: DeBakey and Stanford. According to the Stanford system, dissections involving the ascending aorta are classified as type A, whereas those involving only the descending aorta are classified as type B. The DeBakey system refers to dissections evolving from the ascending aorta and affecting all aortic segments (type I), those affecting only the ascending aorta (type II), and those affecting only the descending aorta (type III) [9]. Cases of aortic dissection involving additional vessels originating from the aorta, such as the subclavian arteries, coronary arteries, and renal arteries, have been reported in the literature [10,11,12,13]. The cause of death is usually blood extravasation in the pericardium, thorax, or abdomen due to the rupture of the aneurysm. Not all cases, however, involve a rupture, and death may result from myocardial ischemia. Several mechanisms have been suggested in these cases, such as the interruption of blood flow to the coronary arteries by the flap, extension of the bulged thrombi in the false lumen into the coronary artery, and a decrease in diastolic pressure and extension of the dissection to the coronary arteries [14,15,16].

## 2. Case Presentation

The case of a 64-year-old male who died suddenly was referred for autopsy to the Department of Forensic Medicine and Toxicology, School of Medicine, National and Kapodistrian University of Athens, as the cause of death was undetermined. His medical history included coronary disease with coronary by-pass surgery at the age of 43, uncontrolled hypertension, diabetes mellitus, and elevated levels of cholesterol. According to his wife, he complained about strong chest pain. During his transfer to the hospital, he lost consciousness and, upon arrival, he was immediately intubated. He was submitted to advanced cardiopulmonary resuscitation, but he finally died.

During the autopsy, aortic dissection was observed along the entire length of the thoracic and abdominal aorta. The dissection of the aorta extended to the common and internal carotid arteries until their entrance into the cranial vault; the subclavian, axillary, brachial, and radial arteries; the superior mesenteric artery; and the common and external iliac arteries (Figure 1).

The pericardium was opened, and the pericardial cavity contained only a small amount of pericardial fluid, without blood. The great vessels were transected about 2 cm above the aortic and pulmonary valves and the heart was examined. The heart was of increased weight—655 g—with a mild left ventricle dilatation (4.5 cm). The thickness of the wall was 1.2 cm for the left ventricle, 1.3 cm for the interventricular septum, and 0.3 cm for the right ventricle. The right coronary artery and the branches of the left coronary artery showed severe atherosclerosis, which caused atheromatic stenosis of the lumen over 75% along the greater part of the vessels. The right coronary artery and the anterior descending and diagonal branch of the left coronary artery were by-passed by the right and left internal thoracic (mammary) arteries, respectively. An old infarction that extended to the anterior wall of the left ventricle and interventricular septum, as well as lesions consistent with recent myocardial ischemia (Figure 2), were observed during gross and histological examination. The aortic dissection was also extended to the right coronary artery and the anterior descending and circumflex branches of the left coronary artery.

Gross and histological examination of lungs revealed pulmonary edema and hemorrhage. The kidneys presented with a granular surface, and histological examination showed nephrosclerosis, findings which are indicative of chronic and severe hypertension. There was no histological evidence of aortitis or connective tissue disease. No other remarkable findings were observed from the rest of the autopsy. No hemothorax, hemopericardium, hemoperitoneum, or any other hemorrhage were found. The death was attributed to myocardial ischemia due to extended aortic dissection.

## 3. Discussion

There are two classification systems for aortic dissection: DeBakey and Stanford; for the purposes of our discussion, we will focus on the Stanford system. Stanford Type A is defined as any dissection that involves the ascending aorta and it necessitates immediate surgical repair, whereas Stanford Type B does not involve the ascending aorta and, in most cases, can be successfully managed with conservative treatment with antihypertensives [9]. Our case is classified as a Stanford Type-A dissection.

Chronic hypertension is by far the most common risk factor for the development of aortic dissection, and is seen in 62–73% of patients with aortic dissection. However, a variety of other risk factors exist, including diseases of the aorta (e.g., bicuspid valve, coarctation, and aneurysm), connective tissue diseases (e.g., Marfan’s and Ehrlers-Danlos syndromes), Turner syndrome, trauma, illicit drug use (such as cocaine), and previous cardiac surgical interventions or catheterizations, which can also cause aortic dissections [1]. Most aortic dissections occur in males with chronic hypertension after the age of 65 years old. Patients with genetic connective tissue disorders and patients with bicuspid aortic valves are at increased risk of aortic dissection at a much younger age (less than 40 years of age) [9].

Aortic dissection typically involves an intimal tear with subsequent exposure of the medial layer to the pulsatile blood flow, the creation of a false lumen between the intima and media or adventitia, and potential propagation proximally or distally. The intimal tear is frequently observed in segments that have sustained the greatest shear stress, such as the right lateral wall (opposite the main pulmonary artery) of the ascending aorta or in the proximal segment of the descending aorta. Propagation can be followed either by aortic rupture or re-entrance into the true lumen through another intimal tear. Aortic rupture leads to exsanguination and death in a short period. Even in the absence of aortic rupture, the mortality rate is very high, especially in cases whereby the dissection extends into the aortic branches and the coronary arteries [17,18].

Peripheral vascular complications have been described in several reports to be present in 21% to 33% of patients presenting with spontaneous aortic dissection in the form of peripheral vessel occlusion from emboli or dissection, with or without occlusion [19,20,21]. Hughes et al., in their study, described peripheral complications from thirty-eight major vessels affected in 18 patients, including the carotid, subclavian, celiac, mesenteric, renal, iliac, and femoral arteries [19].

Since then, a few reports of Stanford A aortic dissections extending to the peripheral vessels have been published. In aortic dissection extending to the arch, the carotid arteries may be involved. Yeh et al. described the case of a 56-year-old woman who was admitted to hospital with vague chest pain and focal neurologic deficits, and she was diagnosed after CT angiography with Stanford A aortic dissection involving the bilateral common carotid arteries [22]. The involvement of the bilateral common carotid arteries, revealed using a Doppler ultrasound, was also reported by Demiryoguran et al. in a 63-year-old female with vertigo as the main symptom [23]. Lee et al. described the case of a 61-year-old man with Stanford A aortic dissection and bilateral common carotid dissection, as well as dissection of the entire length of the aorta down to the right common iliac bifurcation. The left subclavian–axillary, brachiocephalic, and right internal and external carotid arteries were also involved, as depicted in intravenous contrast CT aortic angiography. The man had a history of chronic hypertension and presented with chest pain, vertigo, left facial drooping, and left hemiparesis. The patient underwent surgical repair with a successful outcome [12].

Main abdominal arterial branch involvement has also been reported. Occlusion of the celiac trunk may cause splenic or hepatic infarction with pain and abnormal liver blood test results. Involvement of the mesenteric branches can lead to mesenteric ischemia with nausea, vomiting, abdominal pain, bloody diarrhea, sepsis, and abnormal hepatic and pancreatic enzymes [21]. The renal arteries are very rarely compromised, but when their dissection happens, it leads to ischemia with oliguria, anuria, and abnormal renal blood parameters [24]. Ganduz et al. described the case of a 59-year-old woman with Stanford A dissection, in which the dissection flap seen in the abdominal aorta extended to the celiac artery, the origins of the mesenteric arteries, and the left renal artery. There was also distal extension to the carotid vessels [11].

Fan et al. described the case of a relatively young and healthy man (34 years old) with a history of chronic hypertension and tobacco use, in which extensive Stanford Type-A aortic dissection extended proximally to the carotid arteries and distally to the infra-renal abdominal aorta [10].

Extension of the aortic dissection to the coronary arteries has also been reported and is fatal due to myocardial ischemia, even in the absence of aortic dissection rupture [13,25].

Aortic dissection may prove to be a challenging diagnosis, not only for the clinician but also for the pathologist and/or the forensic doctor. In cases in which rupture has not taken place and there is no macroscopically visible accumulation of blood in the pericardium, thorax, or abdomen, the diagnosis of aortic dissection may not be so obvious. Careful examination not only of the aorta but of its branches, too, may reveal further extension of the dissection to smaller arteries, which is a devastating condition and a serious precipitating factor to the occurrence of death. Although extension to the peripheral vessels has already been reported in the literature, in the autopsy case presented herein, a Stanford A aortic dissection involved a large number of vessels including the common and internal carotid arteries; the subclavian, axillary, brachial, and radial arteries; three coronary arteries; the superior mesenteric artery; and the iliac arteries. It is noteworthy that these findings were present in a man without aortitis or connective tissue disease.

## 4. Conclusions

The case presented herein is rare because of the massive extension of the aortic dissection to the common and internal carotid arteries; the subclavian, axillary, brachial, and radial arteries; three coronary arteries; the superior mesenteric artery; and the iliac arteries, in a man without aortitis or connective tissue disease. The death did not result from the rupture of the aortic dissection, but from myocardial ischemia due to coronary occlusion in combination with hemodynamic disturbance from the stress of the largely extended aortic dissection. Pathologists and forensic doctors should be aware of this rare situation, and meticulous examination not only of the aorta, but also of its branches, should be always performed.

## Figures and Tables

**Figure 1 healthcare-11-00386-f001:**
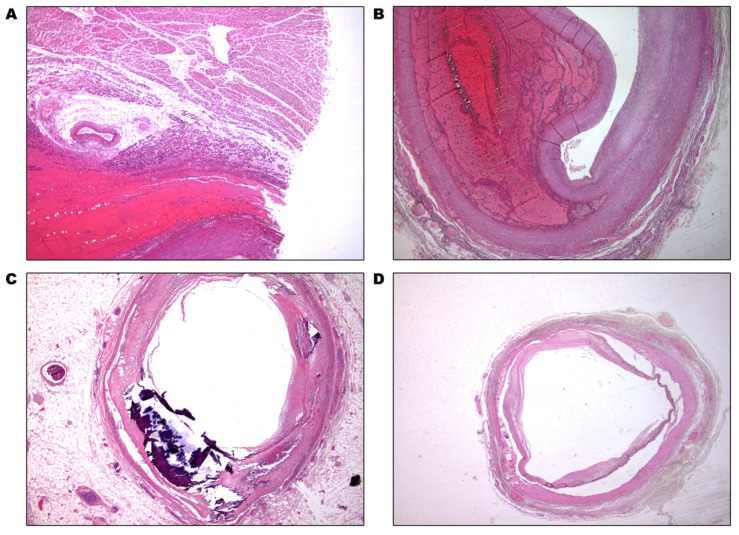
Dissection of (**A**) aorta, (**B**) right common carotid artery, (**C**) right internal carotid artery, and (**D**) right radial artery.

**Figure 2 healthcare-11-00386-f002:**
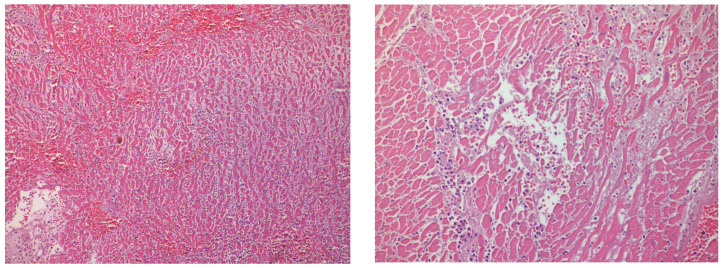
Ischemic changes in the myocardium with edema and blood extravasation.

## Data Availability

Not applicable.

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
