# Peer review of "Extended Stanford Type-A Aortic Dissection with Multivessel Coronary and Peripheral Artery Involvement: An Autopsy Case Report"

_healthcare, 2023, doi:10.3390/healthcare11030386_

Round 1

Reviewer 1 Report

I would like to congratulate the authors on the selection of this novel and the interesting aspect of cardiology. The authors have provided good evidence to support their conclusion with well constructed and meticulously written manuscript. 

However, I would like to bring attention to the following points

1. Please elaborate on the novelty of this case and how it is different from other case reports published in the past

2. In the results section provided results in graphical format can be helpful for readers 

3. In the limitations section more succinct presentation is needed. 

Author Response

We would like to thank the Reviewer for his/her valuable suggestions and for giving us the opportunity to submit a revised version of our manuscript. We appreciate the Reviewer’s insight and we hope that we accommodated most of his/her suggestions in the revised manuscript. All changes have been highlighted in the revised manuscript.

Reviewer 1

  1. Please elaborate on the novelty of this case and how it is different from other case reports published in the past

The following paragraph refers to the novelty of the case report. The sentences that they have been added are highlighted with yellow color:

Aortic dissection may prove to be a challenging diagnosis, not only for the clinician but also for the pathologist and/or the forensic doctor. In cases in which rupture has not taken place and there is no macroscopically visible accumulation of blood in the pericardium, thorax or abdomen does not exist, the diagnosis of aortic dissection may not be so obvious. Careful examination not only of the aorta but of its branches too may reveal the further extension of the dissection to smaller arteries, which is a devastating condition and a serious precipitating factor to the occurrence of death. Although extension to peripheral vessels has already been reported in the literature, in the autopsy case presented herein a Stanford A aortic dissection involved a large number of vessels including common and internal carotid arteries, subclavian, axillary, brachial and radial arteries, three coronary arteries, superior mesenteric artery, and iliac arteries. It is noteworthy that these findings were present in a man without an aortitis or connective tissue disease.

“2. In the results section provided results in graphical format can be helpful for readers 

  1. In the limitations section more succinct presentation is needed

We are not sure about the reviewer’s suggestions. Could we have further explanations?

Reviewer 2 Report

I read the proposed manuscript with great interest. The article is well-written and fits into a very relevant research area. 

However, there are a few aspects that the authors should clarify:

1. Introduction seems to be a little poor of content. In the introduction must be presented the context in which you present a case report. Reader cannot understand what you will describe. If you publish on Healthcare you have to read the section "instructions for authors". At the paragraph "case report", is reported: "Case reports should include a succinct introduction about the general medical condition or relevant symptoms that will be discussed in the case report; ". Of course you cannot follow word by word this instruction, but you must be as detailed as possibile about informations required. 

2. in the same manual you can read: "the case presentation including all of the relevant de-identified demographic and descriptive information about the patient(s), and a description of the symptoms, diagnosis, treatment, and outcome;". You have to follow an ex ante method: in lines 35-37, authors assume the reader already knows the patient is unconscious. I think you can better describe symptoms in a specific pattern: 1. Symptoms 2. vital parameters 3. Instrumental exams.

3. Line 39-42. I think this is a great occasion to show a good example of technical methods of dissection of a heart-great vessels block. Please provide in this report a detailed description of the block and a figure. 

4. Heart's weight is 655 grams, with aorta? please specified. 

5. In this report, the authors write that there is right coronary stenosis but do not specify at what level. So it is not clear whether the stenosis is due to a plaque or whether it is due to a plaque and dissection. Maybe, must be clarify this point. 

6. The authors state that myocardial ischemia due to the extended aortic dissection was the cause of death, since there were no ruptures of the vessels and extensive hemorrhages. I disagree, cause of death in autopsy must be assumed by instrumental, macroscopic or histological data. Now, diagnosis like this doesn't need an autopsy, if you don't provide histological data of a myocardial infarction. Or clinical evidence of STEMI/NSTEMI. Please provide an histological sample in which author demonstrate an acute myocardial infarction or a normal myocardium which lead to a diagnosis of hyper acute-IMA. 

Overall, the results are interesting, some improvements would be necessary.

Author Response

We would like to thank the Reviewer for his/her valuable suggestions and for giving us the opportunity to submit a revised version of our manuscript. We appreciate the Reviewer’ s insight and we hope that we accommodated most of his/her suggestions in the revised manuscript. All changes have been highlighted in the revised manuscript.

Reviewer 2

  1. Introduction seems to be a little poor of content. In the introduction must be presented the context in which you present a case report. Reader cannot understand what you will describe. If you publish on Healthcare you have to read the section "instructions for authors". At the paragraph "case report", is reported: "Case reportsshould include a succinct introduction about the general medical condition or relevant symptoms that will be discussed in the case report; ". Of course you cannot follow word by word this instruction, but you must be as detailed as possibile about informations required. 

The introduction was modified as follows (added text highlighted in yellow)

Aortic dissection is a life-threatening condition that affects 5–30 per 1,000,000 individuals each year (1). Patients may present with pain and shortness of breath, symptoms that mimic other conditions, such as ischemic cardiac disease and perforated gastric ulcer. It is associated with high morbidity and mortality rates and many patients die shortly after the onset of symptoms, before presentation to the emergency department, or before a diagnosis is established (2,3). Historically, many of the early reports regarding acute aortic dissection were based to a great extent on autopsy findings [4]. Since then, several autopsy studies have shown that in a considerable percentage of cases, aortic dissection is first diagnosed during the autopsy, even when patients were under medical care (5-8).

Aortic dissections are classified anatomicaly by two systems: DeBakey and Stanford. According to the Stanford system, dissections involving the ascending aorta are classified as type A, whereas those involving only the descending aorta are classified as type B. The DeBakey system refers to dissections evolving from the ascending aorta and affecting all aortic segments (type I), those affecting only the ascending aorta (type II), and dissections affecting only descending aorta (type III) (9). Cases of aortic dissection involving additionaly other vessels originating from the aorta, such as the subclavian arteries, coronary arteries, and renal arteries, have been reported in the literature (10-13). The cause of death is usually blood extravasation in the pericardium, thorax, or abdomen due to the rupture of the aneurysm. Not all cases, however, involve a rupture, and death may result from myocardial ischemia. Several mechanisms have been suggested in those cases, such as interruption of the blood flow to the coronary arteries by the flap, extension of the bulged thrombi in the false lumen into the coronary artery, decrease in the diastolic pressure and extension of the dissection to the coronary arteries (14-16).

  1. in the same manual you can read: "the case presentation including all of the relevant de-identified demographic and descriptive information about the patient(s), and a description of the symptoms, diagnosis, treatment, and outcome;". You have to follow an ex antemethod: in lines 35-37, authors assume the reader already knows the patient is unconscious. I think you can better describe symptoms in a specific pattern: 1. Symptoms 2. vital parameters 3. Instrumental exams.

The sentence was modified as follows: During his transfer to the hospital, he lost consciousness and upon arrival, he was immediately intubated. He was submitted to advanced cardiopulmonary resuscitation, but he finally died.  

(Vital parameters and instrumental exams were not available)

  1. Line 39-42. I think this is a great occasion to show a good example of technical methods of dissection of a heart-great vessels block. Please provide in this report a detailed description of the block and a figure. 

The following sentences were added:

The pericardium was opened, and the pericardial cavity contained only a small amount of pericardial fluid, without blood.  The great vessels were transected about 2 cm above the aortic and pulmonary valves and the heart was examined.

(The great vessels were transected and the heart-great vessels were not examined as block but separately).

  1. Heart's weight is 655 grams, with aorta? please specified. 

We believe that with the addition of the above text, it is clarified that the weight of the heart was measured after the transection of the great vessels.

  1. In this report, the authors write that there is right coronary stenosis but do not specify at what level. So it is not clear whether the stenosis is due to a plaque or whether it is due to a plaque and dissection. Maybe, must be clarify this point. 

The sentence was modified as follows:

The right coronary artery and the branches of the left coronary artery showed severe atherosclerosis which caused atheromatic stenosis of the lumen over 75% along the greater part of the vessels.

  1. The authors state that myocardial ischemia due to the extended aortic dissection was the cause of death, since there were no ruptures of the vessels and extensive hemorrhages. I disagree, cause of death in autopsy must be assumed by instrumental, macroscopic or histological data. Now, diagnosis like this doesn't need an autopsy, if you don't provide histological data of a myocardial infarction. Or clinical evidence of STEMI/NSTEMI. Please provide an histological sample in which author demonstrate an acute myocardial infarction or a normal myocardium which lead to a diagnosis of hyper acute-IMA. 

Figure 2 where myocardial ischemic changes are shown was added.

Round 2

Reviewer 2 Report

I would like to thank you for following my suggestions and for the excellent presentation. 

Histological specimens are awesome.

 Advise. Maybe, next time you're to be aware to improve your autopsy technique.